# Creating Climate-Resilient Crops by Increasing Drought, Heat, and Salt Tolerance

**DOI:** 10.3390/plants13091238

**Published:** 2024-04-29

**Authors:** Tharanya Sugumar, Guoxin Shen, Jennifer Smith, Hong Zhang

**Affiliations:** 1Department of Biological Sciences, Texas Tech University, Lubbock, TX 79409, USA; tsugumar@ttu.edu (T.S.); jennifer.r.smith@ttu.edu (J.S.); 2Zhejiang Academy of Agricultural Sciences, Hangzhou 310021, China; guoxinshen@gmail.com

**Keywords:** abiotic stress, drought, food security, heat, salinity, climate change

## Abstract

Over the years, the changes in the agriculture industry have been inevitable, considering the need to feed the growing population. As the world population continues to grow, food security has become challenged. Resources such as arable land and freshwater have become scarce due to quick urbanization in developing countries and anthropologic activities; expanding agricultural production areas is not an option. Environmental and climatic factors such as drought, heat, and salt stresses pose serious threats to food production worldwide. Therefore, the need to utilize the remaining arable land and water effectively and efficiently and to maximize the yield to support the increasing food demand has become crucial. It is essential to develop climate-resilient crops that will outperform traditional crops under any abiotic stress conditions such as heat, drought, and salt, as well as these stresses in any combinations. This review provides a glimpse of how plant breeding in agriculture has evolved to overcome the harsh environmental conditions and what the future would be like.

## 1. Negative Effects of Abiotic Stresses

**Impacts of drought stress:** Drought is one of the major hazards in the world affecting agricultural production. A recent report from the Intergovernmental Panel on Climate Change (IPCC) states with high confidence that global warming will lead to more territories becoming affected by agricultural and ecological drought, resulting from insufficient soil moisture that could be intensified by evapotranspiration increase [1]. National Centers for Environmental Information [2] reported that low soil moisture and groundwater levels affect agricultural lands worldwide to a greater extent. It also indicated that drought has caused severe yield loss, leading to increased food prices over the world.

Drought can be classified into four types: meteorological drought caused by less magnitude of precipitation for an extended period of time, agricultural drought referring to the impacts on vegetation when a soil moisture deficit along with high evapotranspiration affects crop yield, hydrological drought that occurs when there is a water deficit in surface or subsurface water reservoirs following meteorological drought, and socio-economic drought that arises when supply cannot meet the demands of economic goods due to the water deficit [3]. The effect of drought stress on plants depends on the severity, duration, and the growth stage of crops at which it occurs. The water deficit typically causes changes in plant morphology such as root length, plant height, total biomass, root/shoot ratio, and photosynthetic pigment content [4]. Leaf rolling, leaf abscission, and yellowing are symptoms observed under severe drought stress. Imbalance between the net photosynthesis and reactive oxygen species (ROS) production due to reduced CO_2_ intake under drought stress results in oxidative damage. Malondialdehyde (MDA), the indicator of lipid peroxidation, increases significantly, indicating the membrane degradation by ROS produced under drought stress [5].

When plants were treated with restricted watering, a significant delay in flowering time was observed among the landraces of common bean [6]. Also, pod number and seed-by-pod reduction caused a significant yield reduction by 50% compared with that of the control treatment. The percentage of yield reduction showed a negative correlation with the percent of relative leaf area expansion, which suggests that, in indeterminate landraces, the resource allocation to pod is higher than to the leaves, improving the yield under drought stress [6]. In rice, flowering is found to be more vulnerable to drought stress, which results in a significant reduction of 23–24% in grain yield and increased spikelet sterility in two cultivars [7]. Chalkiness and chalky kernels were increased by more than 50%, which is caused by the lower accumulation of photosynthates [7]. Drought stress in cultivated cotton varieties resulted in a significant reduction of boll number per plant, and fiber quality was decreased with the increase in micronaire and short fiber percentage [8].

**Impacts of salt stress**: Freshwater availability is becoming scarce due to intensive global freshwater consumption for agriculture, industry, households, public services, and the increased salinity level of surface water in the arid and semiarid regions of the world. Over the past century, the water withdrawal rate exceeded the global population growth rate by more than 1.7 times, where nearly 70% is used only for agriculture. Out of the total precipitation received on earth, only 39% is converted to renewable freshwater resources such as rivers, lakes, and groundwater aquifers. Thus, the overconsumption of water will lead to water scarcity in more than 80% of crop lands in the future [9]. Saltwater intrusion into aquifers due to climate change is a major concern for the salinity increase, which poses a threat to agricultural production by limiting moisture availability to plants [10]. In addition, the water usage has caused coastal land depletion, which has led to a rise in sea levels along with salinity increase in aquifers. Nearly 1.5 million ha of arable land is lost per year due to increased sodicity and salinity, which in turn has reduced the total land availability by 26 million ha [11]. 

Soil salinity is generally measured by the electrical conductivity of a saturated soil paste extract (ECe) that assesses the ability of a solution to conduct electricity, which in turn provides an estimation of the concentration of soluble salts in the soil. Cations, such as sodium (Na^+^), potassium (K^+^), magnesium (Mg^2+^), and calcium (Ca^2+^), and anions such as chloride (Cl^−^), nitrate (NO_3_^−^), and sulfate (SO_4_^2−^) are found in the salts that are commonly present in water and soil solutions. Soils with an ECe of ≥2 dS m^−1^ (at 25 °C) are traditionally considered to be saline soils [12,13,14]. Changes in the weather pattern such as precipitation and temperatures, sea water intrusion, weathering of rocks, and anthropogenic activities such as incorrect agricultural practices like irrational irrigation and the overuse of fertilizer applications cause the salinity of soil to increase. 

Crop tolerance to different soil salinity levels differs depending on the maximum threshold value at which the yield tends to decrease or above which the yield will decrease. Salinity has negative impacts on crop germination and yield, causing morphological, physiological, and biochemical impairments [15]. Leaf wilting, yellowing, curling, and leaves falling are common symptoms observed under salt stress. Depending on the severity scale of 1 to 6, the maximum score for leaf damage in the seedling stage was obtained 24 days after 300 mM of NaCl concentration treatment [16]. Photosynthetic parameters such as chlorophyll (Chl) *a*, *b*, and total content, and photosynthetic rate are reduced and MDA content increases, which are all signs of salt stress [16]. Morphological parameters including plant height, stem and root length, and number of branches and leaves also have a negative relationship with increasing salt concentration [17]. Like other abiotic stresses, flowering and grain-filling stages are more sensitive to salt stress, which directly affects crop yield. Parameters such as flowering time, maturation period, and plant height are reduced significantly in plants grown under high saline conditions [18]. Salt stress in cotton imposes similar negative effects on seed germination and seedling development, biomass, ion homeostasis, and antioxidant activity in susceptible varieties [19,20]. Even though cotton is considered as a salt-tolerant crop with a threshold value of 7.7 dS m^−1^ next to barley, cotton’s growth, boll development, yield, and fiber quality are all reduced by salinity stress [8,21].

**Impacts of heat stress:** According to the recent IPCC report [1], the global temperature is expected to increase by, or by more than, 1.5 °C from 2021 to 2040. This will greatly affect food production worldwide, as the yield of the four major food crops, wheat, rice, maize, and soybean, will be reduced by 6.0%, 3.2%, 7.4%, and 3.1%, respectively [22]. The yield of cotton will be reduced by 10–17% per degree Celsius increasing [23]. Extreme heat will severely affect crop production, as plants are sensitive to changes in temperature during all the stages of their lifetime. From seed germination to seed setting, all stages are sensitive at various degrees. An elevated temperature of 30–35 °C has been shown to significantly reduce the germination rate of most wheat genotypes [24]. Temperatures above 33 °C will cause a yield reduction in rice [25] and affect the seed viability and germination potential during the vegetative stages [26], because heat stress affects three main factors of germination: soil moistures, enzyme activities, and production of hormones. Germination occurs when the outer seed coat breaks open by absorbing water and then the embryo grows by protruding the radical out. Moisture should be sufficient for the seed to imbibe water and trigger germination by softening the seed coat. Hot temperature causes evaporation and reduces the moisture content available for the seed. Enzyme activity is important in the metabolism of starch, proteins, sugar, and lipids, mobilization of nutrients in endosperm, and cell wall breakage of endosperm and seed coat by hydrolytic enzymes [27]. When the temperature increases beyond the threshold level, these enzymes become inactive, affecting the germination process. Viability loss and lower enzyme activities were observed in wheat when the seeds were pretreated at 50 °C [28]. Phytohormones such as abscisic acid (ABA) and gibberellic acid (GA) are involved in the regulation of seed germination, where ABA promotes seed dormancy, whereas GA promotes seed germination [29]. Heat stress decreases GA concentration and increases ABA concentration, thus delaying seed germination [30]. 

At the reproductive stage, heat stress causes detrimental effects to plants and their offsprings. Prolonged heat stress imposed on soybean plants resulted in stunted growth of the fruits, leading to a prolonged reproductive period [31]. It was reported that the seeds obtained from rice plants that were exposed to heat stress during grain filling showed significantly less germination and a lower germination rate compared with the control plants [32], indicating that heat stress can cause epigenetic changes that affect seed germination. Heat stress at anthesis and grain-filling stages reduces the seed setting percentage and photosynthetic rate significantly by affecting the thylakoid membrane integrity, and this is likely due to the effect of heat stress on membrane lipid composition by reducing the desaturase enzyme activities [33,34]. The ROS produced under heat stress also contributes to the damages to membrane lipids by peroxidation [35]. Heat stress can also cause abnormalities in flower morphology, reduced pollen viability and pollen tube growth, poor fertilization, reduced spikelet, and grain number, leading to reduced yield. A recent study in Arabidopsis showed that heat stress has a negative effect on male meiocyte when temperature is increased to 34 °C by leading to irregular spindle structures, defects in chromosomal recombination, and prolonged pachytene/diakinesis phases, which are the probable reason for reduced seed setting [36]. However, the degree of heat stress resistance in plants differs depending on the genotype, the adaptive mechanism they possess, such as morphological modifications, differential gene expression, signaling mechanisms, and physio-chemical changes [37,38]. 

**Impacts of multiple stresses**: In nature, abiotic stresses rarely come along; instead, they come in various combinations or with other stresses such as cold, high light intensity, UV light, and ozone among others. For example, drought with high temperatures, drought with salinity, or all three stresses tend to come together in arid and semiarid regions in the world. The impacts on plants by individual stresses are drastically different from those by multiple stresses, and the effects of different abiotic stresses on plants can be synergistic or antagonistic, which usually lead to much bigger damages to plants [39,40,41,42,43]. Elevated CO_2_ was shown to increase photosynthesis tolerance to heat damage and reduce ROS accumulation [44,45]. Similar effects on photosynthesis and water-use efficiency (WUE) were observed when both heat and drought stress were combined with elevated CO_2_ at the anthesis stage of tolerant and heat-sensitive genotypes, but they were not useful in compensating the yield losses [46]. In a drought-tolerant barley variety, combined drought and salt stresses at the anthesis stage reduced spike length and catalase enzyme activity while the individual stresses alone did not have any effect [47]. 

## 2. The Mechanisms of Abiotic Stress Tolerance in Plants

Plants have evolved stress tolerance mechanisms by which they can adjust themselves to survive under stressful conditions in nature [48,49]. These adjustments are crucial to plants, which integrate complex pathways at morphological, physiological, molecular, and biochemical levels. Under drought stress conditions, roots undergo major changes in their growth and architecture including length, density, radius, and degree of lignification as they are the primary sensors of water scarcity and need to penetrate to lower levels of soil to reach water. Leaves are modified to reduce the transpiration rate and maximize the available water resource. Thick cuticle, high P/S ratio (palisade tissue/spongy tissue) with a thick palisade layer leading to a high rate of photosynthesis and less energy spent to transport CO_2_ between stomata and chloroplast and low specific leaf area, the ratio of leaf area to dry weight, with the adaptive ability of plants in resource scarce environments are considered to be indicators of drought tolerance [50,51]. The development of trichomes and the ratio of trichome to stomata are positively correlated to water deficiency [52]. Heat- and drought-tolerant plants share common features in leaf anatomy such as narrow and thickened leaf surface, resistance to shrinking upon desiccation, lower specific leaf area with a less permeable cuticle or protective boundaries like cuticle, high P/S ratio, and deeper root system [53,54,55].

Proline, which acts as an osmotic stabilizer and radical scavenger that helps to reduce the osmotic potential caused by water deficit and protect membranes and proteins from free radicals, was found to increase significantly under the water deficit condition [56,57]. The accumulation of soluble leaf carbohydrates such as sucrose, fructose, and glucose were found to increase significantly with drought stress, which could be the result of increased starch degradation by amylases. They act as energy sources under reduced photosynthesis, adjust osmotic potential, and act as signaling molecules in stress signaling pathways [58,59]. Mannitol, a sugar alcohol in leaves, also increases substantially in response to drought stress, and it acts as an osmo-protectant of macromolecules against hyperosmolality [5,58].

Salt-tolerant plants exhibit thickened or succulent leaves, a waxy outer layer, extensively suberized apoplastic barriers in roots, and seeds with a rigid seed coat maintaining dormancy in unfavorable conditions [60,61,62,63]. Seeds obtained from the mother plant grown on saline conditions (e.g., 200 mM NaCl) showed significantly higher seed viability, and their morphological parameters such as plant height, stem diameter, total branch length, and reproductive parameters such as flowering branch length, flowering branch ratio, and seed production are better than those seeds whose mother plants were grown under normal conditions [19]. This suggests that the fitness of the progenies can be affected by introducing salinity stress on the mother plants.

A recent study conducted in water dropwort claims that Chl *a* and *b* contents, total Chl content, and carotenoids increase with the increasing NaCl concentration, supporting previous studies in leafy vegetables such as amaranth, sugarbeet, cabbage, and lettuce [17,64]. This suggests that increased levels of chlorophyll and carotenoids in plants under saline conditions might increase the antioxidation capacity in plants, which leads to reduced oxidative damage caused by salt stress. The soluble protein content increase induced by salt stress plays a key role in osmoregulation, by acting as nitrogen reserves [65]. Stress response protein like osmotin and osmotin-like proteins (OLP) were found to be increased under salt stress, which induced proline synthesis and reduced the cell damage by ROS [66]. Brassinolide, a type of phytohormone, is also involved in alleviating salt injury to plants by increasing proline content, antioxidant activity, and maintaining ion homeostasis [67].

Plants also maintain their turgor pressure under salt stress by actively pumping solutes into plant cells, thus increasing the solute potential. A proton electrochemical gradient is created by mainly three proton pumps in plant cells to facilitate the active transport of solutes, namely plasma membrane ATPase (PM ATPase), vacuolar H^+^-ATPase (V-ATPase), and the vacuolar H^+^-pyrophosphatase (H^+^-PPase/V-Ppase). PM ATPase extrudes H^+^ from the cytoplasm of plant cells across the plasma membrane into the extracellular space, thus providing the driving force for the uptake of ions and nutrients such as nitrates and sulfates across the plasma membrane. V-ATPase and V-Ppase acidify the intracellular compartments including vacuolar lumen and the electrochemical gradient created by these pumps is eventually utilized by the proton-coupled antiporters to accumulate ions such as Ca^2+^ and Na^+^. Regardless of the term V-ATPase, these proton pumps are associated with membranes of many secretory organelles such as endoplasmic reticulum (ER), Golgi, coated vesicles, provacuoles, vacuoles, and even the plasma membrane [68]. The V-Ppase pump transports protons from cytosol across the vacuolar membrane actively at the expense of inorganic pyrophosphate (PPi) generated from the biosynthesis of macromolecules such as DNA, RNA, protein, and cellulose. There are two types of V-Ppases in plant cells: type I, the primary V-Ppase that accounts for 10% of the vacuolar membrane proteins, and it requires K^+^ for its optimal enzyme activity and synchronized with the H^+^-ATPase activity on the vacuolar membrane; type II, the secondary V-Ppase that accounts for less than 0.3% of type I in the total membrane fractions, and this proton pump does not require K^+^ and is predominantly localized on the Golgi membrane [69,70]. 

Antioxidant enzymes such as superoxide dismutase (SOD), peroxidase (POD), catalase (CAT), ascorbate peroxidase (APX), monodehydroascorbate reductase (MDHAR), glutathione reductase (GR), and dehydroascorbate reductase (DHAR) are the front-line fighters in the antioxidation defense system that protect cellular components from oxidative damages under drought, salt, and other stressful conditions [71]. The antioxidant activities are found to be higher in heat-tolerant varieties and more effective in providing protection at the initial stages of heat stress and under long-term heat stress [72]. In a study conducted in *Primula minima*, a perennial alpine plant, under combined heat and drought stresses, CAT and glucose-6-phosphate dehydrogenase activities were found to be twofold higher than that under heat stress alone [73]. 

A light-inducible protein-encoding gene *(CPRF-2)* and chlorophyll *a*/*b* binding protein-encoding gene *(CAB)* were found to be differentially upregulated among drought-tolerant barley genotypes, resulting in less chlorophyll reduction and yield loss [74]. Terminal heat stress tolerance in wheat is attributed to the stay-green trait that involves a multi-protein complex, and the genotypes characterized as stay green showed higher Chl content, thousand kernel weight, grain yield, and decreased canopy temperature compared with other genotypes under control and heat stress conditions [75]. 

The maximum net photosynthetic rate of heat-tolerant varieties seems to provide thermal adaptation to plants, which enhances their competitive ability over sensitive plants, as the photosystem (PS) II activity is enhanced by the upregulated expression of the *Orange* gene (*OR*) under heat stress [76]. The *Or* gene encodes an enzyme that regulates phytoene synthase, a rate-limiting enzyme in carotenogenesis. In algae, the *Or* gene along with *PsbP1/2*, that encodes the oxygen-evolving enhancer protein 2 of PS II, were upregulated under heat stress and OR was found to interact with the PsbP1 protein, leading to an increased electron transport rate and photosynthetic efficiency, which enhances the thermotolerance [76]. 

However, there are natural variants of OR proteins with a single amino acid change that could regulate chromoplast number and carotenoid accumulation in plants [77,78]. Even though the OR protein is highly conserved in all plants, one single nucleotide polymorphism converts the 108th amino acid residue Arg into His, resulting in higher carotenoid accumulation compared with the wild-type OR proteins. Site-directed mutagenesis of AtOR^Arg^ into AtOR*^His^* mimics melon’s his variant allele, suggesting that this variant might have acquired an additional function and therefore induces chromoplast formation [79]. It has also been reported that the variant protein interacts with ARC3, a plastid division factor protein, by competing with another protein PARC6, thus regulating the chromoplast number [80]. 

## 3. Creating Climate-Resilient Crops

Over the last 30 years, many genes have drawn the attention of genetic engineers to create transgenic crops that can cope with varying degrees of abiotic stresses and to increase crop yield to the maximum possible under unfavorable conditions. These candidate genes and the gene products are largely classified into two groups: the first group directly involves or has a function in the stress response, such as channel proteins, antioxidant enzymes, chaperone proteins, and osmolytes etc.; the second group regulates stress signaling transduction and expression of the first group of genes, such as transcription factors, signaling molecules, and enzymes in phospholipid metabolism [81]. Currently, various techniques such as sequencing normalized cDNA libraries [82], transcriptome and metabolomic analysis [83,84,85], and computational approaches [86] are being utilized to identify the potential genes that could improve plant performance under environmental stress conditions. 

## 4. Regulation of Gene Expression to Improve Abiotic Stress Tolerance

Nuclear factor Ys (NF-Ys) are transcription factors that bind to the CCAAT box in the promoter regions of their target genes and they form a heterotrimeric complex with three subunits, namely, NF-YA, NF-YB, and NF-YC, which are conserved in all eukaryotes [84]. Recently, many NF-Ys playing key roles in abiotic stress tolerance have been discovered. One such transcription factor, NF-YB1, was overexpressed in Arabidopsis, constitutively driven by the cauliflower mosaic virus 35S promoter (35S), and it confers increased drought tolerance by enhancing water retention potential and increasing photosynthetic rate [87]. When an orthologous transcription factor in maize (ZmNF-B2) was overexpressed, similar results were obtained, with increased yield under field conditions [87]. However, the molecular mechanism behind the increased stress tolerance by NF-Ys remains unclear. 

SIZ1 (SAP and MIZ1 domain-containing ligase 1) is a SUMO E3 ligase that plays a key role in the SUMOylation process in eukaryotes [88]. SUMOylation is a post-translational modification of proteins such as transcriptional factors and chromatin remodeling enzymes, by which a small ubiquitin-like modifier molecule (SUMO) is attached through an enzyme cascade, which helps regulating their activities such as stabilities or sub-cellular localizations [88]. In a recent study, Huang et al. (2023) [89] showed that SUMOylation of the NF-Y complex mediated by SIZ1 is involved in plant thermotolerance. In the SUMOylation process of NF-Y complex, SIZ1 initially interacts with NF-YC10, which then recruits other subunits like NF-YB3 via the SUMO interaction motif (SIM) and NF-YA2 to form a trimer complex for the transcriptional regulation of stress-responsive genes like *HSFA3*. *HSFA3* was shown to be involved in heat stress memory by inducing heat stress memory-related genes directly or via chromatin modifications in Arabidopsis [90]. Transgenic *N. benthaminana* leaves overexpressing *NF-YC10* displayed increased survival rates of seedlings under heat stress compared with the *nf-yc10* mutant, while the *nf-yb3* mutant with mutation in the SIM domain showed reduced association with NF-YC10. Since SIZ1 appears to be involved in all major physiological functions including stress tolerance and NF-YC10 in thermotolerance and photoperiod controlling, the interaction between SIZ1 and different transcriptional factors could be a strategy to improve abiotic stress tolerance. 

In another study by Zhao et al. (2022) [85], transgenic wheat plants overexpressing *TaNF-YA7-5B*, a gene encoding the nuclear factor Y subunit A, were found to have enhanced tolerance to polyethylene glycol (PEG)-induced water deficit. A heterotrimer complex was formed via protein–protein interactions among TaNF-YA7-5B, TaNF-YB2, and TaNF-YC7, which modulates expression of drought-responsive genes. Transgenic Arabidopsis plants overexpressing *CsNF-YC6*, a gene from the tea plant *Camellia sinensis*, showed increased seedling germination and root length when exposed to exogenous ABA, indicating a possible role of *NF-YC* in the ABA-mediated stress response pathway [91].

MYB30, a transcriptional factor regulating salt stress response in Arabidopsis, is SUMOylated by SIZ1 under salt stress [92]. When the SUMOylation site in MYB is mutated via the K283R substitution, an oxidative stress symptom (i.e., ROS accumulation) was evident and the *myb*^K283R^ mutant displayed a sensitive phenotype to salt stress. The cap-binding complex (CBC) is involved in adding 7-methylguanosine (7 mG) cap to the 5′ end of mRNA transcript, which helps transporting mRNA through the nuclear pore, thereby affecting gene-specific effects such as RNA processing and translation [93]. CBC is made up of a small subunit Cbp20 and a larger subunit Cbp80 encoded by *CBP20 (cap-binding protein 20)* and *CBP80*, respectively. The CBC genes are found to be involved in ABA signaling and mutations in the CBC genes led to ABA hypersensitivity and increased drought tolerance [94]. A barley mutant, *hvcbp20.ab*, developed by chemical mutagenesis in *CBP20* was shown to perform better than wild-type plants under water deficit conditions [95]. Similarly, transgenic potato plants with *CBP80*, silenced via an artificial miRNA *(amiR80),* showed improved drought tolerance with morphological adaptations such as increased trichome density and twofold higher stomatal density on the abaxial surface, and compact cuticle with no visible microchannels [96]. 

## 5. Regulation of Post-Translational Modification of Proteins and Enzymes to Improve Abiotic Stress Tolerance

In addition to the roles of SIZ1 in the transcriptional regulation of many genes in stress response pathways, SIZ1 was also shown to be involved in regulating plant growth and developmental processes including cell division and elongation [97], spikelet fertility [98], organ regeneration [99], flowering timing [100], hormonal signaling [101], and heavy metal tolerance [102] via DNA replication, mitosis, DNA repair, nucleo-cytoplasmic transport, and protein stability and interactions [103]. Out of the eight isoforms of SUMO proteins reported in Arabidopsis, SUMO1, 2, 3, and 5 are likely expressed [104]. The non-covalent interaction of SUMO proteins to their target proteins specifically via the SIM sequence is favored by phosphorylated residues near the hydrophobic core of SIM [104]. 

The Arabidopsis *siz1-3* mutants were shown to be hypersensitive to drought with reduced biomass and survival rate, and reduced SUMO protein conjugate levels were detected after drought exposure [105]. This is attributed to the role of SIZ1 in drought response by regulating drought-inducible key genes such as *MYC2*, *ANNAT4*, *KIN1*, and *COI3* [105]. Proteomic analysis of *SIZ1*-overexpressing plants also confirmed that the SUMOylation targets are mostly confined to biotic and abiotic stress responses including drought, heat, salt, immune, and defense response, which makes *SIZ1* the suitable candidate for manipulation in stress-resistant crops [106]. Proving this, *SIZ1*-overexpressing tomato plants remained green with increased chlorophyll content and fresh weight compared with wild-type plants under high heat treatment [107]. When *SIZ1* is mutated in the background of *sos3-1* (a gene encoding a calcium binding protein, salt overly sensitive3 (SOS3), it suppressed the Na^+^ hypersensitivity by decreasing Na^+^ uptake and accumulation while increasing phosphate accumulation. Salicylic acid (SA) level was increased in *siz-1* mutants, and it was proposed that SA and phosphate homeostasis could regulate salt tolerance independently [108]. It was proposed that *SIZ1* negatively affects stomatal closure and drought tolerance via SA accumulation, which poses a question on the interactive nature of SIZ1 under multiple stresses [109]. 

## 6. Maintaining Ion Homeostasis to Improve Abiotic Stress Tolerance

In *Arabidopsis thaliana*, *AVP1* encodes the type I V-PPase and *AVP2/AVPL1* encodes the type II V-PPase that shares 35% similarity in amino acid sequence with AVP1 [110]. There are at least 26 genes that encode the vacuolar H^+^-ATPase that is a multi-subunit complex [111]. In principle, enhancing the expression of either of these proton pumps should result in the increased sequestration of solutes into the vacuole, as the proton chemical gradient established by these two proton pumps drives the sequestration of solutes into vacuole, thereby increasing salt tolerance. Since vacuole is the largest organelle that constitutes 40–90% of the total volume of plant cells, it is responsible for maintaining cell turgor pressure and plant rigidity. Theoretically, increased accumulation of the solutes inside the vacuole will provide plant cells with increased salt tolerance. Gaxiola et al. (2001) [112] demonstrated that by overexpressing *AVP1* in transgenic Arabidopsis would make transgenics plant drought and salt tolerant, which allowed transgenic plants to withstand 150 mM NaCl treatment for 10 days without any stress symptoms, while wild-type plants were all dead. Overexpression of *AVP1* increased the sequestration of cations into the vacuole such as Na^+^ (20 and 40% higher) and K^+^ (39 and 26% higher). The research by Gaxiola et al. (2001) [112] encouraged further studies on *AVP1* overexpression in various plants; all subsequent studies confirmed that overexpression of *AVP1* could substantially improve drought and salt tolerance in transgenic plants [113,114,115,116,117]. 

## 7. Maintaining High Photosynthesis and High Antioxidation Capacity to Improve Abiotic Stress Tolerance

Photosynthesis, being the primary limiting factor of yield and biomass production in plants, is controlled by complex mechanisms [118], among which the key enzyme of the CO_2_ assimilation pathway, ribulose-1,5-bisphosphate carboxylase/oxygenase, or Rubisco, is regulated by Rubisco activase (RCA). Rubisco, upon binding to RCA, undergoes conformational changes in structure, thus removing the sugar phosphate derivative inhibitors that are tightly bound to both active (carbamylated) and inactive (decarbamylated) sites of Rubisco [119,120]. Even though Rubisco is a thermostable enzyme that could remain active up to 50 °C, the heat labile nature of RCA leads to the inactivation of Rubisco, thereby RCA becomes the limiting factor in the assimilation of CO_2_. RCA exists in two forms in most plants, a short (RCA1) and a long (RCA2) isoform, which could be coded by a single gene or separate genes [121] with varying degrees of thermostability. It was found that some plants have thermostable RCA in nature, and this has been explored and is being used widely in genetic engineering. RCA in wild-type rice *Oryza australiensis* [122] and RCA1 β isoform in wheat [123] are such examples. Some studies with overexpression of the RCA gene alone did not result in a significant increase in the photosynthetic rate [124], while other studies showed that co-overexpression of a RCA gene with a small subunit gene of Rubisco (*RBCS*) led to a better result in assimilation of CO_2_ under heat stress conditions [125]. 

Dehydrin (dehydration protein) has received much attention in abiotic stress tolerance in recent years. As a part of the defensive mechanism, organelles such as chloroplast, mitochondria, and peroxisomes produce ROS, which act as stress-responsive signaling molecules. However, excessive accumulation of ROS can lead to cell toxicity and eventually cell death [71]. Therefore, the antioxidant enzymes and proteins that scavenge ROS play a major role in cellular detoxification. Dehydrin belongs to a class of late-embryogenesis abundant (LEA) proteins, and it could serve as a ROS scavenger and a protector of the antioxidant enzymes. Genes encoding dehydrin proteins *SbDhn1* and *SbDhn2* from sorghum were overexpressed in tobacco plants, and the transgenic tobacco plants displayed higher capacity in scavenging ROS and showed enhanced activities of antioxidant enzymes such as SOD, APX, and POX [126]. The *DHN* gene was found to be expressed at higher levels in drought-tolerant varieties [127] and under oxidative stress conditions [128], further supporting the role of the dehydrin protein in abiotic stress conditions. In addition, it was found that DHN (i.e., COR 410) is also involved in leaf rolling response under osmotic stress and is regulated by ABA to ensure better acclimatation under abiotic stress conditions [128]. 

## 8. Co-Overexpression of Two or More Genes to Improve Crop Tolerance to Abiotic Stresses

Although much progress has been made in identifying genes that play important roles in conferring abiotic stress tolerance over the last 30 years, almost none of these genes have been successfully utilized to improve crop yield in field conditions. The failure in translating successful results from laboratory experiments into real gains in crop yield is likely due to the following two reasons: one, the increased tolerance by overexpressing those “stress tolerance” genes is still not powerful enough to overcome the real stress that transgenic plants encounter in the field; two, the stresses that transgenic plants face are too complex that transgenic plants could not handle them, as abiotic stresses usually come in combinations [39,48]. For example, heat stress tends to lead to drought stress in arid and semiarid regions, as heat stress increases transpiration, leading plants to lose water faster; the application of fertilizers in arid and semiarid regions often results in soil salinization, leading to combined drought and salt stress conditions [49,129]. In addition, soil composition, wind-caused mechanical stress, and biotical factors such as bacterial and fungal populations in soil all contribute to the failures that we could not duplicate the successful results from laboratory experiments in field conditions. 

If one can substantially increase tolerance to single stress in transgenic plants, then it might increase crop yield under single-stress conditions. Currently, by overexpressing a single gene to increase salt tolerance, the maximal salt tolerance is approximately 100 mM to 150 mM NaCl in transgenic plants; however, if co-overexpressing two or three genes, it is possible to further increase salt tolerance. Gaxiola et al. (2002) [130] proposed that co-overexpression of *AVP1* and *AtNHX1* would further increase salt tolerance, as AVP1 provides the proton motive force that energizes the activity of the Na^+^/H^+^ antiporter (i.e., AtNHX1). Indeed, when *AVP1* and *AtNHX1* were co-overexpressed in cotton, the salt tolerance level was increased to 250 mM NaCl [116]. However, when *SOS1*, *SOS2*, and *SOS3* were co-overexpressed in various combinations in Arabidopsis, including co-overexpression of these three genes, they failed to further increase salt tolerance in transgenic Arabidopsis plants [131], indicating that co-overexpression of genes in the same pathway or in the same tolerance mechanism might not be effective in generating higher salt tolerance. In contrast, in co-overexpressing genes that function in synergism or in different tolerance mechanisms, it is possible to obtain higher salt tolerance. Because AtNHX1 on vacuolar membrane mediates salt tolerance via Na^+^ sequestration into vacuole [130], while SOS1, the Na^+^/H^+^ antiporter on plasma membrane, mediates salt tolerance via Na^+^ exclusion [132], then co-overexpression of *AtNHX1* and *SOS1* should further increase salt tolerance in transgenic plants. Indeed, Pehilvan et al. (2016) [133] demonstrated that *AtNHX1/SOS1*-co-overexpressing Arabidopsis plants could tolerate a salinity level up to 250 mM NaCl, which was far better than *AtNHX1*-overexpressing and *SOS1*-overexpressing Arabidopsis plants under saline conditions. Recently, Balasubramaniam et al. (2022) [134] showed that the salt tolerance level could be increased to 300 mM NaCl by co-overexpressing *AVP1*, *PP2A-C5*, and *AtCLCc* in Arabidopsis, which further validated the idea that co-overexpression of well-chosen salt-tolerant genes would further increase salt tolerance. In this study, the increased expression of *AVP1* provides more protons that energize secondary antiporters on vacuolar membrane such as AtNHX1, AtCLCa (NO_3_^−^/H^+^ antiporter), and AtCLCc (Cl^-^/H^+^ antiporter), and the increased expression of *PP2A-C5* further activates AtCLCa and AtCLCc, because AtCLCa and AtCLCc are likely the substrates of PP2A-C5 [135], thus the increased Na^+^ sequestration into vacuoles might explain how the highest salt tolerance is achieved (Figure 1).

To increase tolerance to multiple stresses, it is necessary to introduce several genes that function synergistically or mediate stress tolerance via different mechanisms. Over the last 10 years, a few studies were undertaken to simultaneously increase tolerance to heat, drought, and salt stresses, and the results from these studies appear promising. For example, co-overexpression of *AVP1* and *OsSIZ1* in Arabidopsis greatly increased tolerance to single stress of drought, heat, and salinity or in any combinations of these stresses [113], which leads to a significant increase in seed yield than wild-type, *AVP1*-overexpressing, and *OsSIZ1*-overexpressing plants under several abiotic stress conditions. When these genes were introduced into cotton, it was found that *AVP1*/*OsSIZ1*-co-overexpressing cotton plants produced significantly higher fiber yields than wild-type, *AVP1*-overexpressing, and *OsSIZ1*-overexpressing cotton plants under drought, heat, and salt stress conditions in the laboratory as well as in field conditions [136]. Sun et al. (2018) [117] demonstrated that co-overexpression of *AVP1* and *PP2A-C5* in Arabidopsis leads to significantly increased seed yield under combined drought, heat, and salt stress conditions. Subsequently, Cai (2022) [137] showed that *AVP1/PP2A-C5*-co-overexpressing cotton outperformed wild-type cotton under drought and salt stresses in laboratory conditions and produced a higher fiber yield in field conditions. Wijewardane et al. (2020, 2021) [120,138] proved that a heat-tolerant RCA from a desert shrub could be used to improve heat tolerance in transgenic plants, which prompted us to create drought-, heat-, and salt-tolerant cotton by co-overexpressing *AVP1* and *RCA*. Subsequently, we demonstrated that *AVP1/RCA*-co-overexpressing cotton plants produced the highest fiber yield than wild-type and *AVP1*-overexpressing cotton plants under single as well as multiple stress conditions in the laboratory. In field conditions, *AVP1/RCA* co-overexpressing cotton plants produced a fiber yield that was at least 66% higher than WT cotton plants from two years of field trial experiments [139]. Based on these experiments, we believe that by improving crop’s abiotic stress tolerance it is possible to increase crop yield in the near future. A few candidate genes that could be used to substantially increase crop’s tolerance to multiple stresses are provided in Figure 1.

## 9. Use of New Technologies to Improve Crop Stress Tolerance

**Gene editing technology:** A revolutionary technology in plant breeding towards abiotic stress tolerance was the development of genome editing tools such as the clustered regularly interspaced short palindromic repeat (CRISPR)/CRISPR-associated protein 9 (Cas9) technique that accelerated the process of developing tolerant crops [140,141]. The concerns about introducing foreign genes into plants are eliminated in this technique as it precisely edits the target gene at nucleotide level without any transgene intervention [142]. Knockout mutants of diverse genes involved in abiotic stress response created by the CRISPR technique resulted in enhanced tolerance in a range of species such as rice [143], tomato [144], tobacco [145], soybean [146], and wheat [147]. To ensure the gene stability in the modified plants, a lipofection-mediated delivery into protoplasts [148] was developed along with other approaches such as Agrobacterium-mediated transformation [149] and particle bombardment [150]. The modified version of the CRISPR editing, called prime editing, is becoming popular in gene therapy and crop breeding as it precisely alters the target site without the need for donor DNA. Due to these advancements and being not labeled as GMOs, gene editing has greater potential in making the products reach the market quicker than other approaches. 

**Artificial intelligence technology:** With the recent advancement in artificial intelligence (AI) technology, genetic algorithm-based hybrid models have been introduced to predict the stress tolerance index in plants, which will help assess the genotype, taking account of crop production, management strategies, and the prevailing climatic conditions [151]. Given that the crop–stress relationship is dynamic in nature, the AI tools can effectively use morphological and anatomical parameters, especially the root features, to evaluate the tolerance of different genotypes [152]. Deep learning algorithms can be employed in recognizing genes that are differentially expressed under control and stressed conditions by analyzing both temporal and non-temporal data precisely [153]. The advancement allows multispectral sensors and imaging technologies to contribute to modelling AI algorithms, which allows for the non-destructive phenotyping of beneficial traits. For example, relative reflectance indices of combined wavelengths calculated by analyzing each pixel of hyperspectral images of leaves provides accurate quantitative analysis of various stresses [154]. Another sophisticated technique of biosensing miRNA concentrations and digitizing into the degree of stress response was introduced recently, which will reduce most of the laboratory works in the future [155]. This is undoubtedly a great innovation in the abiotic stress research in recent years. 

**Use of miRNAs:** Micro RNAs (miRNAs) are the potential mediators of stress tolerance in plants as they regulate the expression of key genes involved in stress responsive networks. They are 21–23 nt in length and engage in the silencing of the target mRNA expression via post-transcriptional regulation or by mRNA decay [156]. In recent years, several miRNAs have been identified using high throughput sequencing methods, which helped the studying of their role in abiotic stress tolerance (Table 1). 

## 10. Using New Management Strategies and Cultivating Methods to Improve Abiotic Stress Tolerance

**Exogenous applications:** An economic and eco-friendly strategy to make plants resilient to abiotic stresses could be through management practices such as the exogenous applications of phytohormones or growth regulators, seed priming/treatment, or the addition of soil amendments that can alleviate stress impacts. These practices are gaining interest recently due to concerns regarding unlimited use of fertilizers driven by greed for yield increase and their detrimental effects on the environment. For example, ascorbic acid, a metabolite with versatile roles in abiotic stress tolerance, is being referred to as the switch in designing stress-tolerant crops. It has multi-functional properties such as serving as a co-factor for enzymes involved in ABA and GA synthesis, a reducing agent for ROS with high redox potential, inducing SA, JA, and ethylene in pathogen defense, regulating nutrient uptake, and has many positive effects in plant growth and development [167]. Therefore, either the exogenous application of ascorbic acid or enhancing ascorbic acid biosynthesis can alleviate the detrimental effects of abiotic stresses. A virus-induced gene silencing targeted on *GhIMP10D*, a gene encoding the myo-inositol phosphatase that is involved in ascorbic acid synthesis, led to increased sensitivity to alkaline stress in cotton [168], indicating a positive role of ascorbic acid in alkaline stress tolerance. Exogenous application of ascorbic acid was shown to be effective in alleviating the negative effects of heat stress, as in conserving the leaf turgor pressure, decreasing the accumulation of ROS, and increasing nutrient absorption [169,170].

The relative expression of heat shock protein genes was found to be low in ascorbic acid-treated plants, confirming the significant role of ascorbic acid alone in heat stress resistance [171]. The positive effects of ascorbic acid on yield under drought stress as an individual application or in combination with other growth mediators such as proline [172], tocopherol [173], benzyl aminopurine (BAP), moringa leaf extract [174], and chitosan [175] were also reported. 

Another chemical extensively used externally on plants is melatonin that is ubiquitously present in plants and animals and has pleiotropic effects on plants. As a metabolite synthesized from tryptophan, melatonin acts as an antioxidant in ROS defense, regulates cross talk between other phytohormones and ROS, and mediates many developmental processes including germination, circadian rhythm, fruit ripening, and in abiotic stress tolerance. In a recent study, the transcriptomic analysis of common bean plants treated with 100 µM melatonin showed an increased synthesis and metabolism of tryptophan that is involved in flavonoid metabolism, and alleviated the salt stress by affecting the cell wall-related gene expression [176]. Similarly, the antioxidant property of melatonin in conferring salt tolerance was reported in many plants such as fenugreek [177], eggplant [178], wheat [179], mustard [180], maize [181], legumes [182], and drought stress in quinoa [183], and basil [184]. 

**Soil amendments:** Biochar is a modified version of the very old ancient techniques of using carbon black material or charcoal as soil amendments to improve the physio-chemical properties of the soil, thus providing beneficial properties such as water and nutrient retention, reducing the necessity of adding fertilizer, preventing nutrients from leaching, and making the plants more enriched under unfavorable stress conditions, eventually increasing crop yield [185]. Modern biochar, which is inspired from the carbon black material, is formed through a process called pyrolysis, in which biomass rich in carbon is heated to high temperatures with limited oxygen. With the increase in the pyrolysis temperature to 600 °C, porosity, ash content, buffering capacity, and the sorption ability of the biochar were found to be increased and suggested to be the better choice for neutralizing the soil salinity [186]. The surface area and the pore size of the biochar particle is positively correlated with the sorption ability and the population of soil microorganisms inhabiting it [187]. When biochar application is combined with other management practices, the mitigation of abiotic stress effects was efficient. For example, combined osmopriming with CaCl_2_ and biochar amendment resulted in enhanced antioxidant activity and better translocation of carbon reserves at germination and seedling stages, causing increased biomass production in cowpea under salt stress [188]. An application of 3% nano biochar was shown to enhance multiple growth parameters including shoot length, biomass, and relative leaf height under saline conditions by means of increasing photosynthetic pigment content, antioxidant activity, and notably increasing net assimilation rate by 40% and 86% under salinity and drought stress, respectively [189]. Analysis of how biochar induced drought tolerance revealed that by increasing the available and exchangeable K^+^ concentration in the soil, the accumulation of K^+^ in the root sap increases. K^+^, being an essential nutrient required for the survival of the plant and an osmolyte at the early stages of drought stress, is supplied by the biochar [190]. 

**Seed priming/soaking:** Seed priming is a sustainable and cost-effective technique practiced commonly to increase the seed vigor and germination rate in crop cultivation. It can be done by soaking the seeds in inorganic salt solution (halopriming), water (hydropriming), polyethylene glycol-like osmotic solutions (osmo priming), beneficial microorganisms (biopriming), treating with high or low temperatures, and hormonal priming with phytohormones [191]. In addition to these, chemical priming is also performed where different compounds are added to the seeds to enhance growth, development, and the stress tolerance of plants (Table 2). 

## 11. Concluding Remarks

Considering the global food crisis, the need to increase food production has become critical and climate change-caused weather extremes such as drought, heat, and salinity stresses impose additional threats to food security. Finding a solution for these unpredictable and inevitable stress conditions will be the top priority as they become detrimental to plant growth and development. In order to be vigilant about the combined or simultaneous occurrence of these stresses in nature and the complexity of genes, proteins, and their modifications in the adaptive traits against drought, heat, and salt stresses, more studies need to be done in order to decipher the dynamic roles of the network components in plant stress response and tolerance. Better and more efficient ways to deliver transgenes to crop plants such as plant artificial chromosome will be needed in order to deliver more than four or five genes to crop plants. The management strategies need to be more precise in terms of optimization according to environmental changes and sustainability for long-term effectiveness. The emerging AI technology, along with transgenic and gene editing approaches, and new management strategies can be integrated to create climate resilient crops (Figure 2). 

## Figures and Tables

**Figure 1 plants-13-01238-f001:**
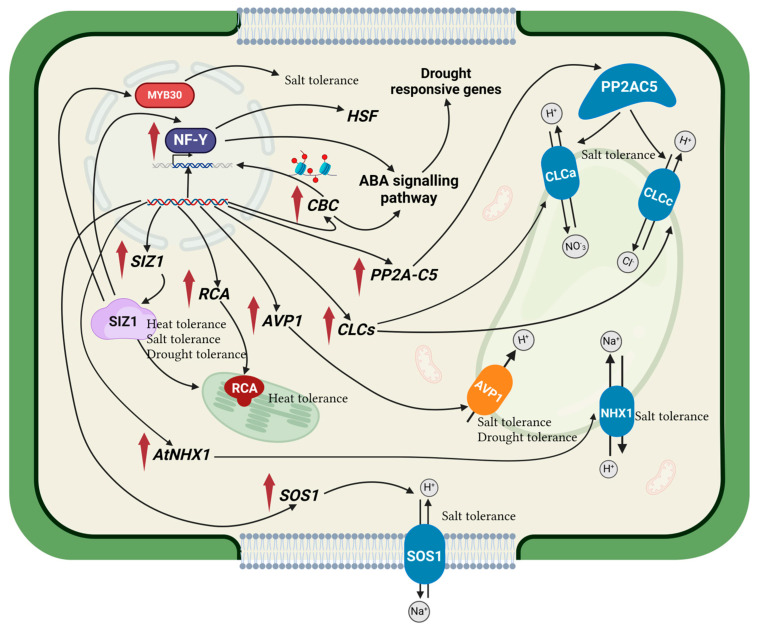
Several candidate genes that could be used to further increase crop tolerance to multiple abiotic stresses when co-overexpressed. Genes such as *AVP1*, *CLCa*, *CLCc*, *SOS1*, *NHX1*, and *PP2A-C5* that are involved in maintaining ion homeostasis can increase salt tolerance when overexpressed. Overexpression of *AVP1* also increases drought tolerance as it stimulates auxin polar transport, leading to larger root system in transgenic plants. *RCA* encodes a heat-stable Rubisco activase (RCA) and it can increase photosynthesis’ heat tolerance when overexpressed in transgenic plants. The SUMO E3 ligase SIZ1 regulates activities of many transcriptional factors and enzymes that are involved in abiotic stress response; when *SIZ1* is overexpressed in transgenic plants, it leads to increased tolerance to drought, heat, and salt stresses. The *CBC* gene encodes a cap-binding complex (CBC) that is involved in ABA signaling and drought tolerance. The nuclear factor Y gene (*NF-Y*) regulates heat and drought stress-responsive genes by increasing their transcriptional activities. This figure was created with BioRender.com.

**Figure 2 plants-13-01238-f002:**
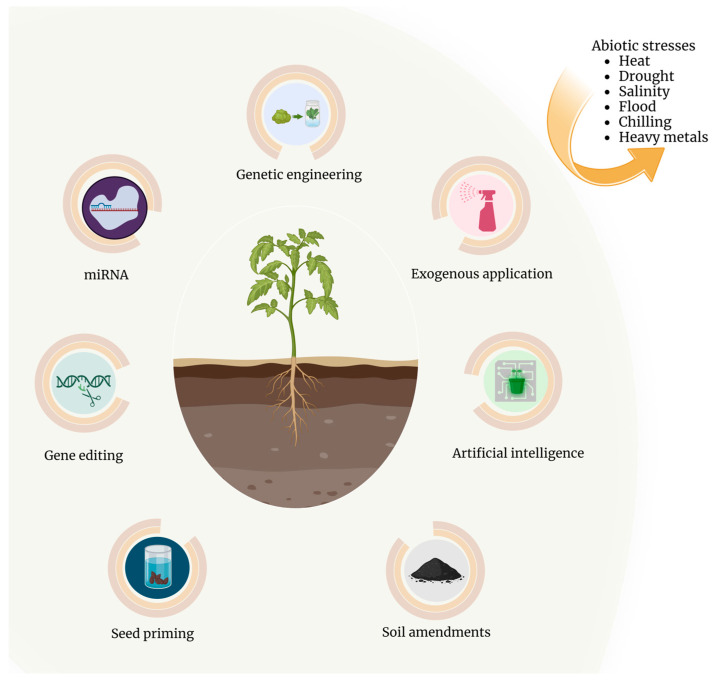
Summary of the possible approaches to create climate resilient crops: These approaches include genetic engineering by overexpressing one or more genes, gene editing technology, use of microRNAs, seed priming, exogenous application of chemicals, use of soil amendments such as biochar, and the use of artificial intelligence. This figure was created with BioRender.com.

**Table 1 plants-13-01238-t001:** List of miRNAs involved in salt, drought, and heat tolerance in plants.

miRNA	Plant Species	Target	Abiotic Stress Tolerance	Effect	Reference
miR396g-5p	*Paeonia ostii*	Assumed to be signal transducer and activator of transcriptional factor (STAT)	Drought	-	[157]
miR396c	*Oryza sativa*	Growth regulating factors (GRF)	Salinity	Overexpression resulted in reduced growth and root length	[158]
miR396-b	Pitaya (*Hylocereus polyrhizus*)	GRF	Salinity	miR396-b was upregulated under salt treatment, thus decreasing GRF gene expression	[159]
miR169	Tomato and *Arabidopsis thaliana*	NF-YA	Heat	Heat shock factors (HSF) induce the expression of miR169 that downregulates *At-NF-YA2/ sly-NF-YA9/10*	[160]
miR396a-5p	Tobacco	GRF-7 regulated osmotic-responsive gene expression	Salt and drought	Overexpression of *miR396a-5p* leads to enhanced leaf RWC, root biomass, antioxidant activity, and survival rate	[161]
miR393	Barley	ABA pathway (NCED1, NCED2, and NCED3)	Drought	Overexpression increased stomatal density and reduced guard cell length	[162]
miR160	*Arabidopsis thaliana*	Auxin response transcription factors (ARF)	Heat	-	[163]
miR408	Wheat (*Triticum aestivum*)	Genes involved in Pi accumulation, signal transduction, microtubule organization, and biochemical synthesis (*TaCP*, *TaMP*, *TaBCP*, *TaFP*, *TaKRP*, *and TaAMP*)	Salt	Overexpression of *TaemiR408* resulted in enhanced growth, biomass, and P accumulation under Pi starvation and salt stress	[164]
miR165/166	*Arabidopsis thaliana*	HSFA1 via *PHABULOSA* (*PHB*)	Heat	-	[165]
miR1861h	*Oryza sativa*	Putative targets include retro transposons, mRNAs encoding transcription factors, methyltransferase, and functional proteins	salt	Overexpression of *miR1861h* resulted in better phenotype under salt conditions	[166]

**Table 2 plants-13-01238-t002:** List of seed priming methods and amendments used in conferring abiotic stress tolerance in plants.

Plant	Amendment	Abiotic Stress	Effect	Concentration	Reference
*Haplophyte* spp.	Melatonin	Salinity	Improved germination and growth	5 and 100 µM MT	[192]
Rice	NaCl, CaCl_2_, KCl, KNO_3_, and H_2_O_2_	Salinity	Increased survival rate	100 mM NaCl, 2.2% CaCl_2_, 2.2% KCl, 2.2% KNO_3_, and 50 mM H_2_O_2_ for 48-h	[193]
Sunflower	H_2_S	Salinity	Maintained ion homeostasis and reduced oxidative damage	0.5 mM	[194]
*Chenopodium quinoa*	CaCl_2_	Salinity	Improved germination and growth, biomass production, and chilling		[195]
Wheat	K_2_SiO_3_	Salinity	Improved seed germination, seedling length	1.5 mM	[196]
Soybean (*Glycine max* L.)	Jasmonic acid	Salinity	Increased net photosynthetic rate, total chlorophyll content and stomatal conductance	60 μM	[197]
Wheat	CaCl_2_	Drought	Improved leaf area, water content in leaf tissue, and yield	1.5%	[198]

## Data Availability

Data are contained within the article.

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
