# Peer review of "Creating Climate-Resilient Crops by Increasing Drought, Heat, and Salt Tolerance"

_plants, 2024, doi:10.3390/plants13091238_

Round 1
Reviewer 1 Report
Comments and Suggestions for Authors
In the article "Creating climate-resilient crops by increasing drought, heat, and salt tolerance" the authors did a good survey of the key points related to plant resilience to climate change.
The only point of climate stress that was not addressed, but perhaps considered to have less agricultural impact, was the flooding process.
The regulation of transcription factor expression in response to stress has also been commented on, but I consider that its effect on various genes and metabolic pathways has not been sufficiently emphasized.
Furthermore, the article provided an adequate approach to mono and multigenic attempts and strategies in the search for stress tolerance, commenting on the difficulty in transposing laboratory data to the cultivation field.
The authors gave particular emphasis to the action of microRNAs, which I consider relevant, but failed to comment on existing data relating the expression of circRNAs in relation to abiotic stresses.
Author Response
In this manuscript, we focus our review on drought, heat, and salt stresses, and we did not intend to be comprehensive. Flooding will be a major issue in the future for some countries like Bangladesh, Indonesia, Malaysia, Philippines, and part of India and China. We will leave this subject for experts in this field to review. Similarly, we did not review those transcriptional factors such as DREBs (CBFs) and NACs that are well known TFs involved in regulating gene expression in abiotic stress response, instead we choose to review those TFs that are involved in SIZ1-mediated abiotic stress response, which is relevant to our research interest. We believe that there will be experts in this special issue of Plants who would be more qualified to review those TFs’ roles in the light of climate changes.
Reviewer 2 Report
Comments and Suggestions for Authors
This is a review article on the effects of important abiotic stresses like drought, salinity and high temperature on plants, diverse response mechanism, and some approaches to create resistant/tolerant varieties of crop species (genetic transformation, gene silencing, and genome editing) or the applications of agricultural practices (exogenous applications of diverse compounds, amendments to try to avoid or decrease the deleterious effects of different stressors). In general, this is a well written and well organized article, which presents to the readers a good overview on the importance of generating improved crop cultivars and the molecular strategies applied until now to produce resistant/tolerant materials; however, because the diversity of the issues, there is not a deep presentation on the basic and integrated complex mechanisms involved in the resistance/tolerance of plants against drought, salinity and heat stresses.
Some minor corrections have been highlighted in yellow in the attached file, which must be accomplished.

Author Response
We have made all changes as requested and thank you for pointing out those problems. Please see the table in the letter to editor that lists all the changes made as you required.
Reviewer 3 Report
Comments and Suggestions for Authors
The first impression already from the title of the review is that it should be very superficial, since the problem of plant resistance to abiotic stresses also as underlying mechanisms are very complex, diverse, and may differ in different species depending on the genetic nature of the plants, ontogenesis stages, the nature of the stressor, etc. However, a difficult task, the review of the problem of creating Climate-Resilient Crops that resist the three main abiotic stressors has been generally successfully solved.
Very briefly but succinctly is described damage caused to plants by three main abiotic stressors, mechanisms to counter them, modern technologies that can be used to remake crops. The authors note that although much progress has been made in identifying genes that play important roles in conferring abiotic stress tolerance over the last 30 years, almost none of these genes have been successfully utilized to improve crop yield in field conditions. However, in our times of rapid development of new technologies, the light at the end of the tunnel is visible and the authors leave hope for the use of a number of the described methods by which Climatic Resilient Crops can be created. These are gene editing technology; artificial intelligence technology; use of miRNAs. Using new management strategies and cultivating methods also could be useful.
To prepare the article, 198 literary sources were used.
Author Response
Thanks for your comments. It is easy to make plants more tolerant to one stress or two stresses in the laboratory, but difficult to make them tolerant to stresses in the field. As you pointed out, tolerance to abiotic stresses are usually multigene traits (usually identified by plant breeders as QTL traits), it is not possible to make crops tolerant by manipulating a single gene via gene transfer technology. More strategies and new technologies will be needed in order to create climate resilient crops. We hope that our review will stimulate more research in this area and we will be able to create crops that can handle the harsh conditions brought by climate changes in the future.